# Characterization of Fluidic-Barrier-Based Particle Generation in Centrifugal Microfluidics

**DOI:** 10.3390/mi13060881

**Published:** 2022-05-31

**Authors:** Masoud Madadelahi, Javid Azimi-Boulali, Marc Madou, Sergio Omar Martinez-Chapa

**Affiliations:** 1School of Engineering and Sciences, Tecnológico de Monterrey, Ave. Eugenio Garza Sada 2501, Monterrey 64849, NL, Mexico; jazimib1@binghamton.edu; 2Department of Mechanical Engineering, Isfahan University of Technology, Isfahan 84156-83111, Iran; 3Department of Mechanical Engineering, Binghamton University, Binghamton, NY 13902, USA; 4Department of Mechanical and Aerospace Engineering, University of California Irvine, Irvine, CA 92697, USA; mmadou@uci.edu

**Keywords:** fluidic barrier, centrifugal microfluidics, droplet generation, alginate microparticle, three-phase model

## Abstract

The fluidic barrier in centrifugal microfluidic platforms is a newly introduced concept for making multiple emulsions and microparticles. In this study, we focused on particle generation application to better characterize this method. Because the phenomenon is too fast to be captured experimentally, we employ theoretical models to show how liquid polymeric droplets pass a fluidic barrier before crosslinking. We explain how secondary flows evolve and mix the fluids within the droplets. From an experimental point of view, the effect of different parameters, such as the barrier length, source channel width, and rotational speed, on the particles’ size and aspect ratio are investigated. It is demonstrated that the barrier length does not affect the particle’s ultimate velocity. Unlike conventional air gaps, the barrier length does not significantly affect the aspect ratio of the produced microparticles. Eventually, we broaden this concept to two source fluids and study the importance of source channel geometry, barrier length, and rotational speed in generating two-fluid droplets.

## 1. Introduction

In recent decades, considerable efforts have been made regarding particle production and handling in microfluidic systems for a broad range of applications ranging from water treatment [1] to cell study and clinical diagnostics [2,3]. The advantages of using microfluidic systems include the low sample and reagent consumption, efficient heat and mass transfer, coexistence of different fluid phases, improved detection limits, compact size, shorter processing time, high-throughput capability, portability of the instrumentation, enhanced operational flexibility, and reduced costs by integrating a full set of operations on a single chip [4,5,6,7,8]. Droplet microfluidics offers exquisite simultaneous control over multiple fluids by allowing precise tuning of both the geometric characteristics and the compositions of the droplets used to synthesize microparticles. Taking advantage of microfluidic techniques, microparticles with diverse morphologies, controlled size, and specific functionalities can be produced [9,10]. 

Polymeric microparticles have emerged as advanced functional materials for a wide range of biomedical applications. Particles made up of natural polymers, such as alginate, chitosan, and gelatin, have found a wide range of applications in the food, cosmetic, biomedicine, agriculture, and pharmaceutical industries due to their abundance, biodegradability, low toxicity, high binding capacity to specific chemical species, and biocompatibility, and their ability to adsorb or release molecules in response to external stimuli. They have been used extensively in drug delivery [11], e.g., encapsulation of islet cells capable of releasing insulin [12,13]; biosensors [14] and actuators [15]; scaffolds for cell culture in tissue engineering [16,17,18]; sorbents [19]; and encapsulation of enzymes and cells [20,21]. Different structures of microparticles are paving the way to a wide variety of applications. Multicore particles with high encapsulation efficiency can be produced with well-defined compositions and structures allowing for controlled release of the encapsulated materials [22,23]. 

Different techniques can be used to make microparticles. Some pumpless microfluidic systems work with gravity, capillary forces, etc., which are not usually high-throughput approaches [24,25]. On the contrary, many microfluidic systems work with pumping units. Syringe pumps are well-known in microfluidic systems, and provide acceptable accuracy of flow rates. However, many of these expensive external pumping systems are unsuitable for highly viscous fluids. They usually require complex interconnects and long tubing, and waste precious samples and reagents. In addition, they are slow, exhibit pulsatile and unsteady flow behavior at low flow rates, and are bulky, and are hence not suitable for point of care (POC) applications [6]. By comparison, internal pumps such as electrohydrodynamic (EHD) [26,27] and magnetohydrodynamic (MHD) [28,29] micropumps can provide pulse-free flow and accurate control of the flow rate. However, they need electrically conductive media to operate, which inevitably generate heat [30]. In contrast, centrifugal pumping is pulse-free, does not require conducting media, and is free of parasitic heat induction. It is biologically friendly, capable of processing highly viscous fluids, inexpensive, and easy to multiplex [6]. Typically, nozzle dispense methods are used for the production of microparticles or beads on centrifugal microfluidics [13,21,31,32]. In this approach, the centrifugal force pushes the fluid through a nozzle and forms tiny droplets at the nozzle tip. Then, droplets enter a continuous phase to crosslink and solidify. The superiority of centrifugal pumping primarily lies in two factors: first, the strong centrifugal force can process highly viscous fluids; second, the strong centrifugal force can pinch off the droplets at the early stages of formation at the tip of the nozzles, resulting in smaller droplets. Therefore, generating smaller droplets does not necessitate the use of extremely small nozzles, which have fabrication challenges and difficulty in transferring cells and biological samples through narrow channels [31]. 

Nozzle clogging is a common issue when the channel or nozzle tip directly contacts the crosslinking solution. To avoid cross contamination and nozzle clogging, an air gap can be used [13,21,31,32]. The air gap size has a significant effect on the morphology of the particles. Previous results show that the greater the distance that droplets fly, the larger deformation they undergo [33]. This is due to the effect of air stress during the traveling time of the droplets. Moreover, higher concentrations of the dispenser phase show smaller deformation because of the viscosity effects. Cheng et al. observed that too-small air gaps also cause the deformation of microbeads and suggested the nozzle position should be at a certain optimal distance from the continuous phase for generating uniformly shaped microbeads [34]. Moreover, it is reported in the works using air gaps that the size of the dispenser nozzle has a direct relationship with the size of generated particles, i.e., the bigger the dispenser, the bigger the particles [13,21,34]. 

Our previous works introduced the concept of a fluidic barrier with applications in multiple emulsion and microparticle generation, and compared this method with other available droplet generation methods [10]. Fluidic barriers can be utilized to separate reactive chemical components, such as a polymer and its crosslinker liquid. Here, we focus solely on microparticle generation application to characterize different controlling parameters and present new details of this phenomenon. We also broaden the concept to include highly viscous working fluids, high rotational speeds, and droplet generation of two source fluids. Using two source fluids can help the automation of the process. Figure 1 illustrates the fluidic system. The gray fluid shows the crosslinker, and the yellow fluid shows the oil that separates the polymer and the crosslinker. Because the centrifugal force depends on the fluid’s density, the crosslinker should be denser than the oil. Two viscous fluids (red and blue) are loaded in separate source reservoirs. Since they have a higher density than their medium, they make a droplet and, while crossing the fluidic barrier, mix before solidification and accumulate in the final reservoir. We show that transient secondary flows within the droplets produce good mixing. We study the importance of the fluidic barrier length, the rotational speed, and the source channels’ width. Establishing the relative importance of these different parameters will help scientists significantly in designing centrifugal microfluidic devices using fluidic barriers. 

## 2. Theory and Governing Equations

On a centrifugal microfluidic platform, different forces play significant roles on the path of the microparticles. Centrifugal and coriolis forces are defined as follows:

(1)$${\vec{F}}_{\mathrm{centrifugal}}=\frac{{\pi \;d}_{p}{}^{3}(\rho_{p}-\rho_{f})}{6}\vec{\omega }\times \left(\vec{\omega }\times \vec{r}\right)$$(2)$${\vec{F}}_{\mathrm{coriolis}}=-\frac{{\pi \;d}_{p}{}^{3}(\rho_{p}-\rho_{f})}{3}\vec{\omega }\times \vec{v_{p}}$$
where $\vec{\omega }$ is the rotational velocity vector, $\vec{r}$ is the radial position vector, $d_{p}$ is the particle’s diameter, $\vec{v_{p}}$ is the linear velocity vector of particle, and $\rho_{f}$ and $\rho_{p}$ are the densities of fluid (893 kg/m^3^ for the crosslinker, 1000 kg/m^3^ for water, and 870 kg/m^3^ for mineral oil) and particle, respectively. 

Yet another important force that acts on each microparticle is the drag force. The drag force is calculated using the Oseen correlation as follows:(3)$${\vec{F}}_{\mathrm{drag}}=-\frac{1}{\tau_{p}}m_{p}\left(\vec{v_{f}}-\vec{v_{p}}\right)$$
(4)τp=4ρpdp23μCDRer
(5)CD=24Rer(1+316Rer)
(6)Rer=ρ|vf→−vp→|dpμ
where μ is the viscosity of fluid (0.0015 Pa.s for the crosslinker, 0.001 Pa.s for water and 0.017 Pa.s for mineral oil), and $\vec{v_{f}}$ is the velocity vector of the fluid. Added mass (Equation (7)) and gravity (Equation (8)) forces also need to be included. Added mass is the force generated when the fluid rushes to open the way for the particles while filling the gap behind them. The particle’s velocity in Equation (7) must be considered relative to the surrounding fluid velocity.
(7)$${\vec{F}}_{\mathrm{added}\;\mathrm{mass}}=-\frac{1}{12}{\pi \rho }_{f}d_{p}{}^{3}\frac{d\vec{{v_{p}}_{r}}}{\mathrm{dt}}$$
(8)$${\vec{F}}_{\mathrm{gravity}}=\frac{{\pi \;d}_{p}{}^{3}(\rho_{p}-\rho_{f})}{6}\vec{g}$$

All of the forces mentioned above need to be combined in Equation (9) to calculate the particle velocity $\dot{(\vec{x_{p}}})$. $m_{p}$, which is the particles’ mass, can be calculated using the experimental particles’ diameter and density.
(9)$$m_{p}\ddot{\vec{x_{p}}}=\sum^{\;}\;\vec{F}$$

In addition to the overall speed of microparticles, we need to investigate the phenomena occurring when a droplet crosses a fluidic barrier. Since we are dealing with three different phases in this study, we need a multiphase model for the detailed theoretical investigation of phenomena in each phase. Here, phases are defined as different immiscible materials. We used the volume of fluid (VOF) model for three different incompressible phases. In this model, volume fraction (α) is defined as the ratio of the volume of each phase to the total volume of each cell. Different phase interfaces are tracked by solving the continuity equation (Equation (10)) for the volume fraction [35].
(10)$$\frac{\partial }{\partial t}\left(\rho_{x}\alpha_{x}\right)+\nabla .\left(\rho_{x}\alpha_{x}{\vec{v}}_{x}\right)=0$$
where x is the index showing different phases. This equation is solved for phases 1 and 2. For the calculation of the volume fraction of phase 3, we use Equation (11) as a required constraint.
(11)$$\overset{3}{\underset{x=1}{\sum }}\alpha_{x}=1$$

A single momentum equation is solved (Equation (12)), and the calculated velocity is shared among all different phases in each cell.
(12)$$\frac{\partial }{\partial t}\left(\rho \vec{v}\right)+\nabla .\left(\rho \vec{v}\vec{v}\right)=-\nabla p+\nabla .\left[\mu \left(\nabla \vec{v}+\nabla {\vec{v}}^{T}\right)\right]+\vec{F}$$
where p is the pressure and F is the surface tension force (N/m^3^), which is calculated as follows: (13)$$\vec{F}=\sigma \kappa \nabla \alpha =\;\sigma \nabla .\left(\hat{n}\right)\nabla \alpha$$
where σ is the surface tension coefficient (assumed as 0.049 N/m for oil phase and 0.072 N/m for water phase), and κ is the curvature (1/m), defined as the divergence of the unit normal vector $\hat{n}$ of the interface between the phases. For the wall adjacent cells, $\hat{n}$ is considered as: (14)$$\hat{n}={\hat{n\;}}_{w}{\cos\theta }_{w}+{\hat{T\;}}_{w}{\;\sin\theta }_{w}$$
where $\theta_{w}$ is the contact (assumed 90° in this case), ${\hat{n\;}}_{w}$ is the unit normal vector of the wall surface, and ${\hat{T\;}}_{w}$ is the unit tangential vector of the wall surface. In all cells away from the walls, $\hat{n}$ is calculated based on the gradient of the volume fraction as:(15)$$\hat{n}=\frac{\vec{n}}{\left|n\right|}=\frac{\nabla \alpha }{\left|\nabla \alpha \right|}$$

All of the required properties in the above equations, such as the density, are calculated using a volume fraction averaging:(16)$$\rho =\overset{3}{\underset{y=1}{\sum }}\alpha_{y}\rho_{y}$$

## 3. Materials and Method

### 3.1. Material Preparation

For biomedical applications, biopolymer alginate is a commonly used material, with previous test results showing its long-term biocompatibility [36,37]. In our previous investigation, we used Na-alginate at low concentrations [10]. Here, we focused on the same material 6% (*w*/*w*), as a highly viscous source fluid. Mineral oil was used as the fluidic barrier. No surfactants were added to the oil phase to help the instability of the oil layer surrounding the Na-alginate cores and expose them to the crosslinker more easily. For the crosslinker solution, we made a CaCl_2_ solution 6% (*w*/*w*). Since we needed a density difference to push the Na-alginate droplets radially outward into the CaCl_2_ solution, we had to dilute the crosslinker down to 50% using ethyl alcohol. All chemicals were purchased from Sigma-Aldrich. For the fabrication of microfluidic devices, the polymethyl methacrylate (PMMA) sheets were purchased from Simbalux, and the pressure sensitive adhesive (PSA) was obtained from 3M.

### 3.2. Fabrication of Microfluidic Devices

All the microfluidic devices in this study were made of three PSA (black color in Figure 2A) and four PMMA (yellow color in Figure 2A) layers. According to Figure 2A, the second and third PMMA layers, and the first and last PSA layers, have a similar design to include all the elements except the source channels. The source channels are only considered on the middle PSA layer (Figure 2B). This structure helps to have the central channels in the middle height of the device so that the generated droplets do not touch the top or bottom PMMA layers. For the fabrication of the microfluidic devices, we cut the patterns on the PSA layers (thickness = 50 µm) using a cutter plotter machine (Graphtec CE6000). We used a precise laser cutter unit (Glowforge plus) to cut the 2400 µm thick PMMA sheets. Many different devices were fabricated, varying the number of source reservoirs or the structure of the source channels. Figure 2C depicts one of the fabricated devices.

### 3.3. Experimental Procedure

The channels and reservoirs of the fabricated devices were filled with the prepared solutions, as shown in Figure 1. The outlets in the final reservoir were sealed by adhesive tape so that the fluidic barrier and the crosslinker stayed within the channels while the device was rotating. Then, the device was mounted on an electric motor controlled by a computer. All the macroscale photos were taken utilizing a synced digital stroboscope (DT-2350PA Digital Stroboscope). All bright-field and fluorescence microscope images were taken off disc using an inverted microscope (Carl Zeiss). During particle generation experiments, some particles can stick together poorly, depending on the centrifugal force and the high number of particles in chambers. In these cases, we waved the samples for 15 seconds using a vortex (DLAB) to separate the particles.

### 3.4. Modeling Methodology

Two different simulations were performed in this study. The first was a finite-volume-based model. Due to the high speed of the centrifugal platforms and the non-continuity of the stroboscope pulses, it was not possible to capture all the details continuously and accurately. Hence, a verified model was needed as a substitute. We used a VOF method and only focused on a concise time frame when a droplet crosses a fluidic barrier. As the verification of this model, the droplet generation phenomenon in a benchmark T-junction microfluidic device was reproduced. We considered different droplet sizes to measure the dimensionless droplet size and distance between the droplets and compared these values with the experimental observations of Tice et al. [38]. Appendix A) shows an excellent agreement between the model and experimental data. In the next step, according to Figure 3A, the geometry and grids (a total number of 5231 after a grid independency test) were generated. As shown, due to very high computational needs, we had to consider this model in 2D. Three immiscible fluids were assigned to each domain. The red circle shows the droplet in phase 1 (water phase), the orange medium shows the oil phase in phase 2, and the yellow rectangle shows the phase 3 fluid (water phase). We used the quadratic upstream interpolation for convective kinetics (Quick) method for the discretization of Equation (12), and the pressure-implicit with splitting of operators (PISO) scheme for momentum–pressure coupling. Furthermore, a first-order implicit method was considered for the time steps. The second simulation was a finite-element model to solve the governing equations mentioned in Section 2 to calculate microparticles’ speed in the centrifugal platform. A similar model was used for another centrifugal device in our previous study and verified by experimental observations [10]. In Figure 3B, we depict the computational domain discretized using 271,893 grids after grid independence tests. For solving the governing equations, a generalized minimal residual (GMRES) method was used.

### 3.5. Data Analysis

All the error bars in this study show the standard error (Equation (17)), where SD is the standard deviation and n is the sample number. The sample number is considered fifteen in all cases.
(17)$$\mathrm{SE}=\frac{\mathrm{SD}}{\sqrt{n}}$$

Fluorescence intensity profiles were obtained from the grayscale images using ImageJ software. These values are reported in dimensionless diagrams where the axes are non-dimensionalized by the particles’ average fluorescence intensity and diameter in each case.

## 4. Results and Discussion

### 4.1. What Happens When a Droplet Crosses a Fluidic Barrier?

We investigate here what happens when a droplet crosses the border of the fluidic barrier. According to the results, when a droplet touches the border of the fluidic barrier, it does not cross it spontaneously. It sticks to the phase interface for a while, and then a super-fast reciprocating process occurs. In Figure 4, we show how the droplet crosses the border of a fluidic barrier; the water phase (phase 1) droplet is shown as a red circle, and the solid black line is the border of the oil phase (phase 2) liquid. Here, the phase 2 liquid is located at the bottom, and the phase 3 liquid is located on top of the interface. As the droplet touches the solid line at t = 0.015 s, one tiny droplet is trapped on top of the main droplet. This is a tiny droplet of phase 2 liquid carried with the red droplet. This droplet remains in the red droplet’s centerline and is carried with that in our model. If this droplet leaves the red droplet’s centerline for any unbalanced condition, it will go back and merge with the oil phase fluidic barrier due to the lower density. Similar to the mentioned droplet, two more tiny oil droplets detach at t = 0.021 s once the red droplet is more clearly in the phase 3 liquid. However, since they have a lower density than the phase 3 liquid, they go back towards the fluidic barrier and merge quickly due to the centrifugal force (Equation (1)). The orange arrows in Figure 4 show these droplets. The yellow arrows on the right section of each image show the moving direction of the red droplet based on a rotating coordinate system, and the yellow circles demonstrate the times at which the red droplet stops moving and the moving direction changes. The droplet velocity inverts its direction at times of 0.039, 0.06, 0.084, and 0.108 s. For the first reciprocating phenomenon at t = 0.039 s, the red droplet has the minimum displacement from the fluid’s interface, and for the second time at t = 0.084 s, this displacement increases. Once the red droplet crosses the interface at t = 0.144 s, the interface shows a quick vibration until t = 0.156 s (the orange arrows in Figure 4). When the droplet is separated, it continues to move forward within the fluidic barrier. 

When the droplet passes through the interface quickly, fluid flow patterns change substantially. In Figure 5, we depict some selected instant fluid patterns inside the droplet and the surrounding fluids. Times associated with particular flow patterns are written on the upper-left side in pink, and a white dashed line shows the liquid phase borders. In addition to the arrows showing the flow patterns, the color here corresponds to the velocity magnitude of the fluid. As shown, there are two symmetrical vortices at t = 0.009 s, before the droplet touches the interface. According to our model, as the droplet touches the interface, the maximum velocity increases initially (as a case in point, see t = 0.021 s). These local high-velocity regions are related to the rapid movement of tiny droplets due to the centrifugal force, as shown in Figure 4. Afterward, the number and position of the vortices around the droplet change, which are marked by the yellow dashed lines. Furthermore, we also notice that when the droplet meets the interface, some tiny vortices appear at the interface itself (see t = 0.039 and 0.042 s). Finally, as shown in the last image at t = 0.165 s, when the droplet crosses the interface, the two circulating vortices return to their initial state (before the droplet touched the interface).

Now that we know precisely how a droplet crosses a fluidic barrier, we need to also find the importance level of different parameters in this phenomenon. These parameters, discussed in the following sections, can help scientists more easily and frequently design and utilize fluidic barriers in their devices.

### 4.2. The Importance of Barrier Length and Source Channel Width on Particles Velocity

In Figure 6A, we represent the velocity contours and the particle trajectories within the fluid for a rotational frequency of 50 Hz and a fluidic barrier length of 3 cm. Although some vortices are made in each phase due to the rotational platform, the velocity magnitude of the fluid particles is small. According to this model, under the same conditions, the fluidic barrier length does not affect the velocity of the particles when they pass the fluidic barrier. In Figure 6B, we depict the particle’s velocity versus the radial distance from the source channel. The only difference between the fluidic barrier lengths of 1, 2, and 3 cm for a rotating speed of 50 Hz is the position of the velocity ramp due to the location of the fluidic barrier interface. We know that this sudden increase is due to the sudden decrease in drag force, which helps particles accelerate within the fluid. The effect of source channel width on the generated particle’s velocity at f = 50 Hz and with a fluid barrier length of 1 cm, is shown in Figure 6C. In this figure, small, medium, and big channel sizes correspond to the widths of 300, 600, and 900 µm, respectively. As the channel width increases, the velocity and acceleration magnitude of the particles increase but its behavior after crossing the fluidic barrier interface is almost linear. The same behavior was observed for the other fluidic barrier lengths of 2 and 3 cm.

### 4.3. The Importance of Fluidic Farrier Length and Source Channel Width on Particles Characteristics

As two of the most significant parameters, we investigated the effect of the fluidic barrier length and source channel widths on the size and shape of the microparticles. According to Figure 7, three different fluidic barrier lengths of 1, 2, and 3 cm are investigated for three different source channel widths at f = 50 Hz. Small, medium, and big channel sizes correspond to 300, 600, and 900 µm, respectively. The mean diameter and aspect ratio are defined as Equations (18) and (19), respectively.
(18)$$\mathrm{Mean}\;\mathrm{diameter}=\sqrt{4\times \frac{\mathrm{Area}\;\mathrm{of}\;\mathrm{the}\;\mathrm{structure}}{\pi }}$$
(19)$$\mathrm{Aspect}\;\mathrm{ratio}=\frac{\mathrm{longest}\;\mathrm{diameter}\;\mathrm{of}\;\mathrm{the}\;\mathrm{structure}}{\mathrm{shortest}\;\mathrm{diameter}\;\mathrm{of}\;\mathrm{the}\;\mathrm{structure}}$$

According to Figure 7, the fluidic barrier length affects the generated particles’ size: the shorter the fluidic barrier length, the bigger the generated particles. This effect is higher when we use smaller source channels. According to Figure 7, if we triple the fluidic barrier length (from 1 to 3 cm), the size of particles generated by the small source channel decreases by 17%, whereas this value for the big source channel is only 1.45%. Figure 7 shows that the fluidic barrier length does not significantly affect the generated particles’ aspect ratio. By comparison, the width of the source microchannel controls both the size and aspect ratio. According to Figure 7, as the width of the microchannel increases, bigger microparticles are generated. This behavior, which was anticipated but not quantified in our previous works, is due to the higher flow rate of fluids through wide channels. This relation between the channel width and particle size was almost linear for the conditions of this experiment. It was seen that the aspect ratio slightly increased by increasing the source channel width. The reason is that, as shown in Figure 6, the velocity and acceleration of the generated polymeric droplets increase when we use a bigger source channel. When a bigger (i.e., faster) droplet smashes on the fluidic barrier–crosslinker interface, its deformation is higher than that of smaller (i.e., slower) droplets. Moreover, the deformation of particles when they accumulate with high acceleration is more when they are big. As a case in point, for the big source channel, Figure 6 showed that the velocity of droplets when they enter the crosslinker is almost 100% more than that for the small source channel. This results in a 13.6% increase in the aspect ratio, as depicted in Figure 7. 

### 4.4. The Importance of Rotational Frequency on Particles Size and Aspect Ratio

Here, the effect of the rotational frequency on the size and aspect ratio of microparticles is investigated. As the rotational speed increases, the size of microparticles decreases. This decrease is not linear and is smaller at higher rotational speeds. According to Figure 8, it is observed that this behavior is independent of source channel width and is similar for both the small and big source channels. Moreover, we did not see any meaningful differences in the aspect ratio of the generated microparticles over different rotational frequencies between 20 and 50 Hz.

### 4.5. The Capability of Using More Than One Source Fluid

This section focuses on using two separate source fluids and investigating the effect of the source channel geometry and the barrier length on the microparticle generation. According to Figure 9A–C, three different types of source channel connections were considered: Type-A, Type-B and Type-C (all other dimensions are similar to Figure 2B). Parameters such as source channel width (380 µm) and intersection angle (19.21°) are constant, and only the position at which the two fluids are in contact with each other is different. In Type-A, Type-B, and Type-C devices, two different source channels join at the end, in the middle, and at the beginning of the two source channels, respectively. According to our experiments, Type-A was not a suitable configuration for making microparticles using two different source fluids. As shown in Figure 9D, both green and black source fluids made separate droplets at all different rotational speeds, and they were not able to join each other. A thin oil layer in the middle of two droplets forced each droplet to be generated independently of the other. Differently to Type-A devices, Type-B and Type-C configurations were able to produce single joint droplets of both source fluids and had similar results (Figure 9E).

In the next step, in order to investigate the mixing of source fluids, Rhodamine-6G florescent fluid was used as the source fluid 1, and DI water was used as the source fluid 2. In Figure 10, the bright-field and fluorescence images are shown at f = 50 Hz. Moreover, we examined the effect of fluidic barrier length by investigating three different fluidic barrier lengths of 1, 2, and 3 cm. In order to quantify the mixing phenomena due to the secondary vortices, the fluorescence intensity was measured along a circular path within each particle, as shown in a yellow circle in Figure 10. This circle has a radius equal to one-half of the particle’s radius (r = R/2, where R is the particle radius). This reference length starts from the left point, and is nondimensionalized by the total length of the path. The measured intensities were normalized by the average fluorescence intensity of the corresponding particle. $\left(\mathrm{relative}\;\mathrm{intensity}\;\mathrm{variation}=\frac{\mathrm{measured}\;\mathrm{fluorescence}\;\mathrm{intensity}\;-\;\mathrm{average}\;\mathrm{fluorescence}\;\mathrm{intensity}}{\mathrm{average}\;\mathrm{fluorescence}\;\mathrm{intensity}}\times 100\right)$. The relative intensity variation versus the normalized reference length is plotted for all different samples. As shown in Figure 10 for the three samples, the fluorescence signal exists along the reference curve, and the maximum deviation of the signal is less than 15%. This observation was seen in all samples. It proves that the fluorescence fluid is mixed with the DI water in all particles. In addition, the fluidic barrier length did not play any role in the final particle’s size and aspect ratio when we used two source fluids. In some of the experiments, if a large number of particles in the chamber are collected, they may stick together poorly (like the particles in Figure 10 for the fluidic barrier length of 3 cm). This is more likely to happen at high rotational frequencies due to higher centrifugal force. Then, we can use a 15-second vibration on a vortex device if we want to separate all of the particles.

Finally, the effect of rotational frequency on the mixing of the two source fluids was investigated. We increased the rotational frequency from 30 to 50 Hz (the maximum motor operating frequency in the set-up) to establish the effects on fluid mixing. Some examples of the bright-field and fluorescence images are shown in Figure 11 using the fluidic barrier length of 2 cm. Relative intensity variation versus the normalized reference length for three samples is shown in the last row in Figure 11. It was seen that, in all generated particles, the fluorescence intensity had a maximum variation of less than 15% compared to the average fluorescence intensity of each particle. This shows that good mixing of two source fluids occurs, even at different rotational frequencies (30 to 50 Hz) due to the secondary internal vortices.

## 5. Conclusions

In this research, we focused on details of using highly viscous working fluids in fluidic barriers in centrifugal microfluidic platforms. For this purpose, two different theoretical models were introduced. One of these models was a three-phase model to capture the detailed sequential snapshots of what happens when a droplet enters a fluidic barrier. To the best of our knowledge, this is the first three-phase model established for centrifugal microfluidics. It was shown that droplets cross the fluidic barrier interface after a high-speed reciprocating behavior at the fluid’s interface. Furthermore, during this process, different secondary flows were captured and detailed. The other model was for the calculation of the generated microparticles’ speed. We showed that the fluidic barrier length is not a critical parameter for the final speed of the microparticles. Our experiments showed that the size and aspect ratio of the generated microparticles increase by increasing the source microchannel width. In addition, we demonstrated that, unlike air gaps utilized in the previous works, the length of the fluidic barrier is not an important parameter for the aspect ratio of the microparticles. However, it slightly affects the size of the microparticles. To further expand the applicability of fluidic barriers in centrifugal platforms, we designed and fabricated configurations using two source fluids. We showed that two source fluids mix perfectly due to the mentioned secondary flows within the generated droplets. It was also shown that this perfect mixing is independent of the fluidic barrier length and rotational speed, and can be utilized in high-throughput microparticle generation. Using separate source chambers can help us to increase the automation of centrifugal microfluidics.

## Figures and Tables

**Figure 1 micromachines-13-00881-f001:**
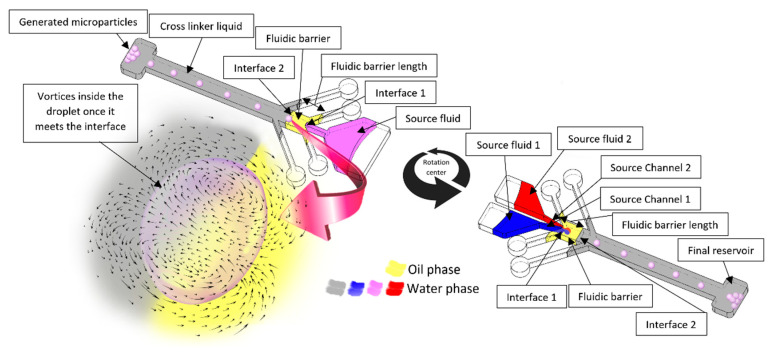
Schematic of centrifugal microfluidic device with one (on the **left**) and two (on the **right**) source fluids and different important parameters.

**Figure 2 micromachines-13-00881-f002:**
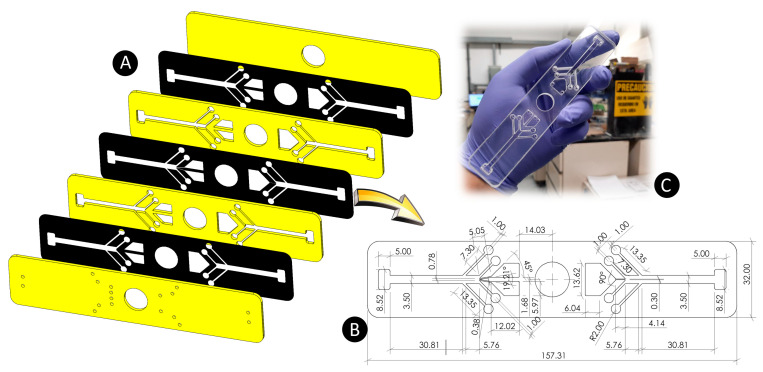
(**A**) Different layers of the microfluidic device, (**B**) the dimensions (mm), and (**C**) one of the fabricated devices. The three black layers represent the PSA, and the four yellow sheets show the PMMA.

**Figure 3 micromachines-13-00881-f003:**
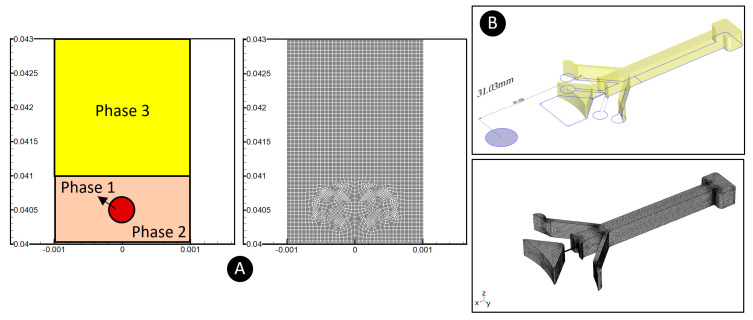
(**A**) The geometry and grids used for the three-phase modeling. The red circle shows the droplet in the water phase (phase 1), the orange medium shows the oil phase fluidic barrier (phase 2), and the yellow rectangle shows the other water phase (phase 3). (**B**) The geometry and grids used for the particle tracing model.

**Figure 4 micromachines-13-00881-f004:**
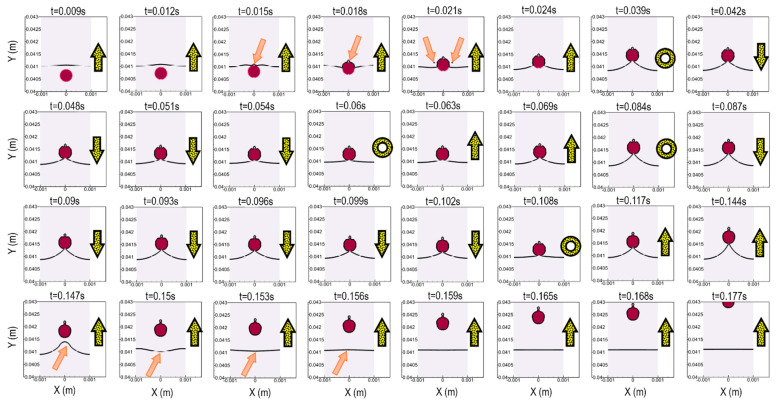
The behavior of a droplet when it crosses the border of a fluidic barrier; the solid black line shows the fluidic barrier edge. Yellow arrows refer to the direction of motion of the droplet, yellow circles show the instance at which the droplet stops and its moving direction reverses, and the orange arrows refer to important details that should be noticed.

**Figure 5 micromachines-13-00881-f005:**
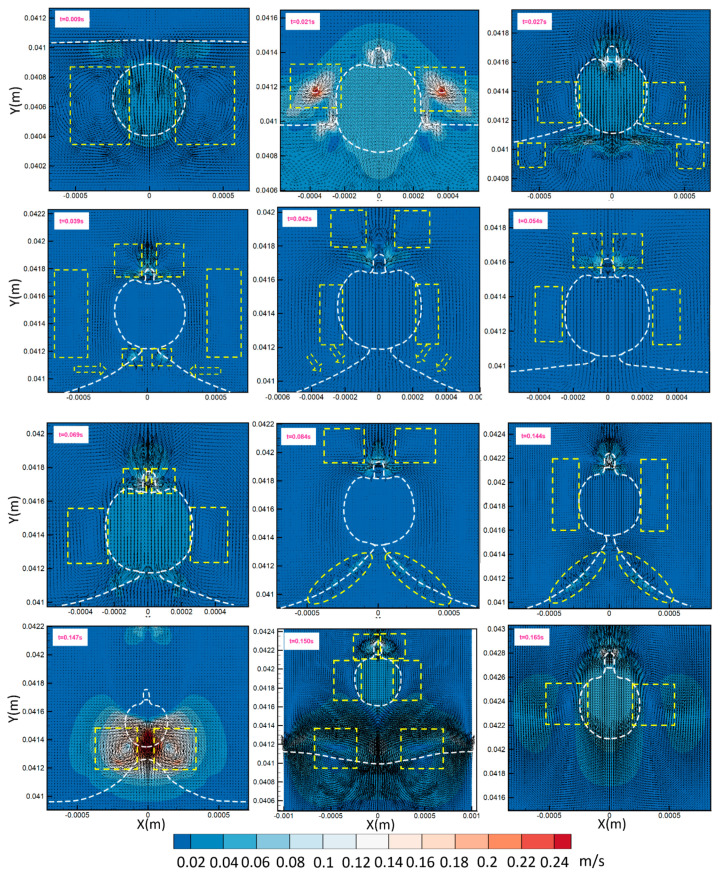
Fluid flow velocity vector (arrows) and velocity magnitude (colored contour) when a droplet crosses a fluidic barrier. The time of each image is shown in the top-left by a pink text. The yellow and white dashed lines indicate the vortices and liquid interface, respectively.

**Figure 6 micromachines-13-00881-f006:**
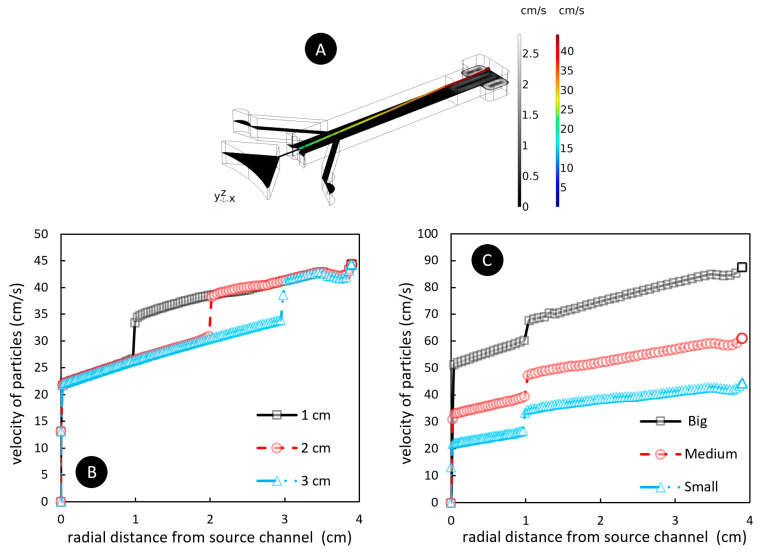
(**A**) The fluid velocity contour inside the microfluidic device, the particle trajectory, and magnitude at f = 50 Hz, using a channel width of 300 µm and fluidic barrier length of 3 cm. (**B**) Particle velocity vs. radial distance from the source channel for different fluidic barriers lengths of 1, 2, and 3 cm at f = 50 Hz. (**C**) Particle velocity vs. radial distance from the source channel for different widths of the source channel at f = 50 Hz. Here, the fluidic barrier length is 1 cm. Small, medium, and big channel sizes correspond to 300, 600, and 900 µm, respectively.

**Figure 7 micromachines-13-00881-f007:**
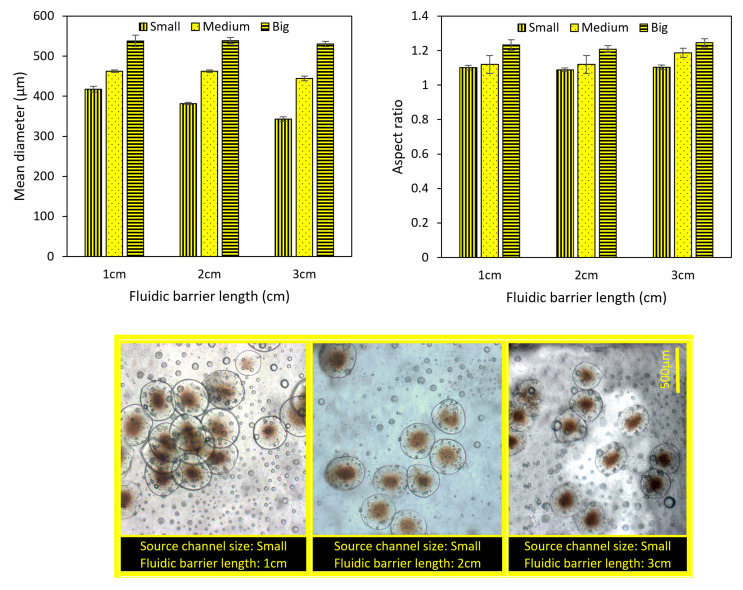
The size and aspect ratio (mean ± standard error) of microparticles with different fluidic barrier lengths and source channel widths (f = 50 Hz). Scale bar is 500 µm.

**Figure 8 micromachines-13-00881-f008:**
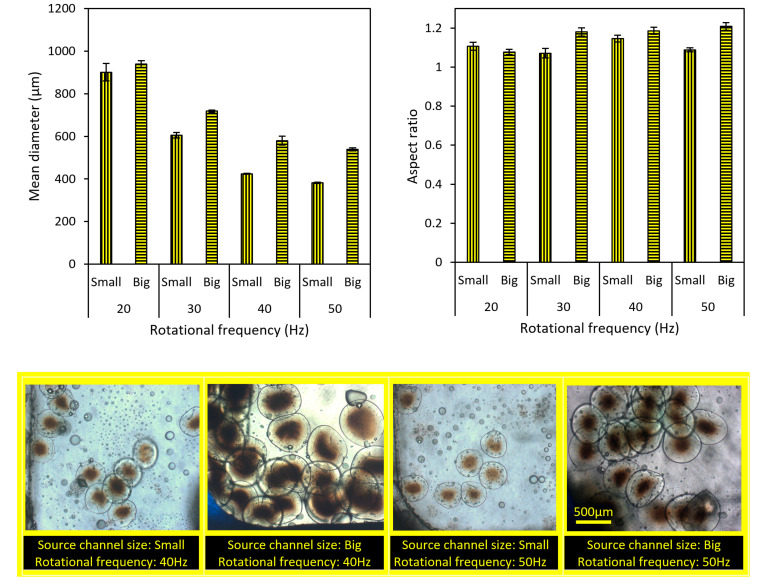
Microparticles’ size and aspect ratio (mean ± standard error) at different rotational frequencies. The fluidic barrier length is considered to be a constant (=2 cm). Scale bar is 500 µm.

**Figure 9 micromachines-13-00881-f009:**
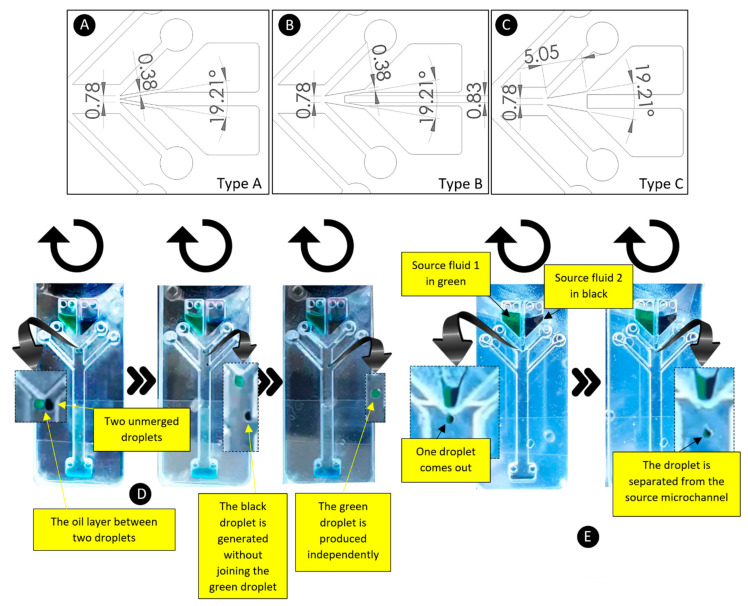
(**A–C**) Three different configurations of the joining source microchannels. The lengths are in millimeters. (**D**) The phenomenon happens using joining channel Type-A. (**E**) The phenomenon happens using joining channel Type-B. This behavior was also observed in Type-C.

**Figure 10 micromachines-13-00881-f010:**
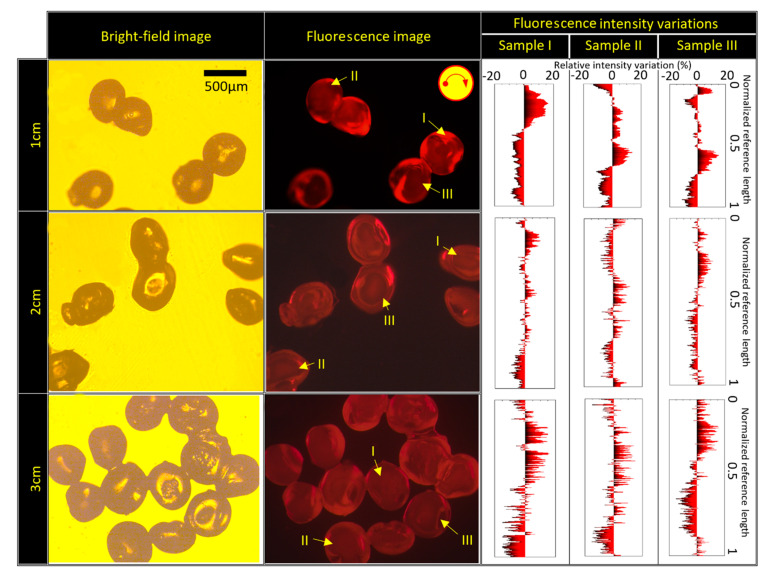
Fluorescence and bright-field images of the generated microparticles using two source fluids with three different fluidic lengths of 1, 2, and 3 cm at f = 50 Hz. Scale bar is 500 µm.

**Figure 11 micromachines-13-00881-f011:**
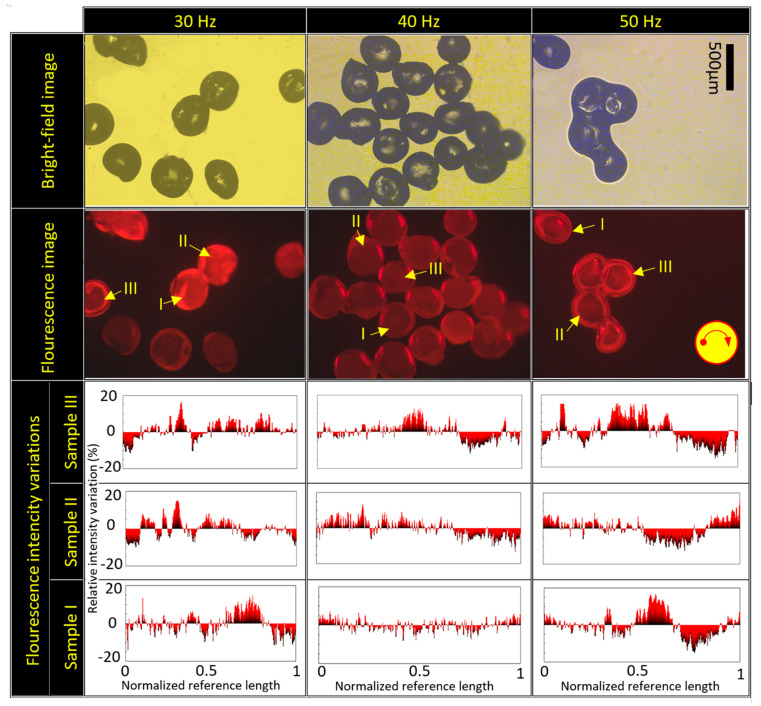
Fluorescence and bright-field images of the generated microparticles at 30, 40, and 50 Hz using the fluidic barrier length of 2 cm. Scale bar is 500 µm.

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
