# Peer review of "Characterization of Fluidic-Barrier-Based Particle Generation in Centrifugal Microfluidics"

_micromachines, 2022, doi:10.3390/mi13060881_

Round 1

Reviewer 1 Report

The presented work seems interesting, but I would suggest making more clear significance of the work. Apart of that I have only some editorial comments. First, I really wonder if any of authors actually read the manuscript, I do not mean "scan" it. 

Figure 11. Fluoresce and bright-field images of the generated microparticles at 30 Hz, 40 Hz, and 50 421 Hz using the fluidic barrier length of 2 cm. 

So what is "Fluoresce"? and on Y Axis, there is not "Fluoresce" but another unusual expression called in the manuscript "Flourescence"

Some other typos are in the manuscript. 

Then there are few more issues, such as ways of introducing abbreviations:

"PMMA (Polymethyl methacrylate)" should read as "polymethyl  methacrylate (PMMA)" without capitalization of P in "Polymethyl"

No reason for capitalization of "E" in "Ethyl alcohol."

"900 microns" That unit of lengths has correct name "micrometer" as "micro" is a slang and it is abbreviated as "µm". 

"t=0.039s" should read as "t=0.039 s"

Reviewer 2 Report

Madadelahi et al. reported using highly viscous fluids in fluidic barriers on centrifugal microfluidic platforms. The authors described different theoretical models with experimental support from detailed sequential snapshots of what happens when a droplet enters a fluidic barrier. There is also a good amount of system optimization by varying the parameters including microchannel width, fluidic barrier length etc. The main methods and results are of practical interest, and the logic/structure of this paper is clear, which can be published after minor revisions:

1.     This study used Na-alginate at high concentration and the author claims this to be a highly-viscous source fluid. Has this been tested by viscometer? The authors reported that the low concentration Na-alginate at low concentrations has been reported previously [8]. How does this study compare to that study? What is the difference when switching between highly-viscus fluid with low viscous fluid?    

2.     Please standardize the dimension unit in the manuscript. In some places, ‘mm’ was in use (such as in Figure 2) but in some other places ‘cm’ was in use. When describing the channel sizes, microns were used, may consider to standardize them all, such as ‘μm’.

3.     In figure 7, there is no scale bar on the fluid barrier length of 1 cm and 2cm image. Same for figure 8. In addition, there is a lack of information on the generated particle sizes relative to the fluidic barrier length. The authors concluded that ‘the shorter the fluidic barrier length, the bigger the generated particles’. But what is the size exactly? Please refer to figures 7 and 8.

4.     On page 12, “This relation between the channel width and particle size was almost linear for the conditions of this experiment.” How did the author get to this conclusion?

5.     Suggesting putting the exact channel size dimensions rather than saying small, big, etc.

6.     In figure 9a, please specify the units of the dimensions

7.     Scale bar of the bright field and fluorescence images in Figure 11.
